# Exploring Interference Issues in the Case of n25 Band Implementation for 5G/LTE Direct-to-Device NTN Services

**DOI:** 10.3390/s24041297

**Published:** 2024-02-17

**Authors:** Alexander Pastukh, Valery Tikhvinskiy, Evgeny Devyatkin

**Affiliations:** 1Radio Research and Development Institute, 105064 Moscow, Russia; vtniir@mail.ru (V.T.); deugene@list.ru (E.D.); 2Institute of Radio and Information Systems (IRIS), 1010 Vienna, Austria; 3International Information Technologies University (IITU), Almaty 050000, Kazakhstan

**Keywords:** D2D, 5G, LTE, NTN, direct-to-device, Starlink, MSS, interference analysis, n25, n256, 3GPP

## Abstract

This paper delves into an interference analysis, focusing on the forthcoming Starlink Generation 2 satellites, stated to operate within the 1990–1995 MHz frequency band. The aim is to assess the potential interference from this Starlink system to the satellite receivers of mobile satellite systems (MSSs), which are set to function within the 1980–2010 MHz range, and satellite receivers of the NTN systems, which are planned to operate in the n256 bands, defined by the 3GPP specifications. Through simulation-based evaluations, both single-entry and aggregate interference levels from Starlink to MSSs and NTN systems are comprehensively explored. To estimate the interference impact, several protection criteria were used. The study is in line with the Recommendations of International Telecommunication Union (ITU-R) and common approaches that are used when performing compatibility studies between satellite systems. The findings of this study demonstrate the feasibility of utilizing the n25 band for NTN direct-to-device services.

## 1. Introduction

In the dynamic landscape of contemporary broadband communications, the pursuit of higher data rates aligns seamlessly with the quest for widespread connectivity. This becomes particularly crucial as more tourists explore remote natural areas, where traditional communication infrastructure is either absent or severely limited. The challenge of providing mobile services in such areas persists, and current satellite solutions, though capable, often demand costly and cumbersome user equipment, discouraging a substantial user base and relegating these systems to niche applications.

A promising solution to overcome this challenge involves delivering broadband satellite services directly to regular handsets, specifically unmodified smartphones. This approach has proven successful in transforming niche applications into mainstream phenomena, as exemplified by the integration of GPS functionality into widely adopted consumer gadgets like smartphones and tablets. Supporting LTE/NR satellite services with regular handsets could potentially provide a viable solution to access broadband services in remote areas.

However, implementing direct-to-device (D2D) satellite systems in terrestrial spectrum bands poses significant challenges, especially when dealing with unmodified smartphones. These challenges encompass interference issues, Doppler shifts, which affect signal frequency and phase, causing distortion and reception problems, and delays complicating the realization of Hybrid Automatic Repeat Request (HARQ) protocols, which are crucial for reducing the Bit Error Rate (BER) in channels, and the inherent limitations of smartphones’ relatively low power, leading to constraints in link budget and throughput. Regulatory challenges also arise due to the operation of these systems in terrestrial spectrum bands, falling outside traditional satellite service allocations.

This study delves into the multifaceted issue of interference in D2D satellite systems, exploring potential problems that may arise when implementing such systems. A notable challenge is the necessity to utilize terrestrial spectrum bands for satellite services, which were not originally intended for mobile satellite services. Some companies, like AST SpaceMobile and Lynk, have ventured into this domain, launching test satellites capable of providing broadband services in UHF terrestrial mobile bands. Although utilizing UHF bands offers advantages due to their excellent propagation characteristics and favorable link budgets, it comes with the constraint of requiring larger antennas, a challenge imposed by payload mass limitations [1].

To address the antenna size limitation, one potential solution is to operate in higher terrestrial spectrum bands. SpaceX and T-Mobile’s collaboration plans to launch second-generation Starlink satellites serving regular handsets within the n25 band, specifically 1910–1915 MHz (Earth-to-space) and 1990–1995 MHz (space-to-Earth). However, this frequency band presents a challenge, as the space-to-Earth link overlaps with the mobile satellite service (MSS) S-band, using the 1980–2010 MHz (Earth-to-space) and 2170–2200 MHz (space-to-Earth) bands. This overlap introduces the potential for reverse space-to-space link interference, a complex issue due to radio waves’ propagation in a vacuum with minimal attenuation. Additionally, the S-band, defined as the n256 band in 3GPP for NTN networks, is supported by future smartphones, creating an overlap with other potential D2D systems built on future NTN networks.

This paper presents an interference analysis conducted between second-generation Starlink satellites and MSS and NTN systems, focusing on the frequency band 1990–1995 MHz. It specifically considers typical MSSs, which are based on ITU filings from various MSS networks and designed to offer voice calls, IoT services, and data transfer services in the S-band. The study also explores the potential interference with NTN systems in the n256 band based on 3GPP specifications. The results will clarify whether the n25 band is suitable for providing direct-to-device services or if interference concerns must be addressed.

## 2. Literature Review

During the World Radiocommunication Conference 2023 (WRC-23), mew agenda item for future conference in 2027 was opened, under this agenda item ITU-R should study possible allocations for MSS that utilizes International Mobile Telecommunication (IMT) user equipment in the terrestrial cellular frequency bands with the 694–2700 MHz range. The IMT based radio interfaces include UMTS, LTE, 5G and future generations of cellular communications. This agenda item essentially will study possibility of using unmodified handsets based on LTE, 5G and future generations within the terrestrial bands of the cellular operators. However, these studies haven’t yet began and will last for the next 4 years until the WRC-27. At national level some countries have already began to update their national frequency regulations to allow satellite systems operate in the terrestrial band [2], however these updates do not eliminate some interference issues.

Thus, presently, there is a limited body of research focusing on interference issues in satellite systems deployed in the terrestrial spectrum, especially in the context of implementing device-to-device (D2D) services. While a few studies have touched upon this subject, the depth of technical analysis remains somewhat limited. This study places emphasis on investigating interference scenarios between Non-terrestrial (NTN) satellite systems and other mobile satellite service (MSS) systems, particularly within the n255 and n256 frequency bands [1].

In the 3GPP TR 38.863 report, titled “Non-Terrestrial Networks (NTN) Related RF and Co-Existence Aspects”, the primary focus revolves around frequency allocations, aligning with the ITU-R Radio Regulations (RR), and encompasses several compatibility studies [3]. One published article delves into the compatibility between satellite and terrestrial segments of 5G networks in adjacent frequency channels [4]. Another publication [5] outlines the simulation methodology for interactions between Non-Geostationary Satellite Orbit (NGSO) and Geostationary Satellite Orbit (GSO) systems. Additionally, there is work discussing the potential of hybrid satellite/terrestrial networks, which includes considerations for traffic sharing between terrestrial and satellite Radio Access Networks (RAN) [6].

This research, exploring the use of 3GPP technology for satellite communication, underscores the importance of conducting interference analyses, highlighting the risks that may arise when the same spectrum is utilized for both terrestrial and satellite systems [7]. Furthermore, Ref. [8] provides insights into potential approaches for dynamic spectrum sharing between terrestrial and non-terrestrial networks in the context of 5G services and beyond.

In summary, while numerous papers acknowledge the challenge of interference when employing the terrestrial spectrum for satellite connectivity, most have refrained from conducting in-depth technical analyses, primarily raising awareness of the issue. Consequently, there is a need for more comprehensive studies that can offer a deeper understanding of the extent of the problem, which is important for correct decision-making processes when deploying D2D services in terrestrial bands.

## 3. Simulation Parameters and Scenarios

In 2021, 3GPP unveiled Technical Report TR 38.821. This report provides a methodology and includes example parameters that enable the simulation of satellite-based non-terrestrial networks (NTN) offering 5G services. Its primary goal is to delineate the essential features and adaptations needed for the New Radio (NR) protocol to operate effectively within non-terrestrial networks, with a particular emphasis on satellite access [9].

Following the finalization of Release 17, several new frequency bands were introduced for the NTN segment of NR technologies. Notably, the n255 band encompasses a pair of frequency bands, 1626.5–1660.5 MHz (Earth-to-space) and 1525–1559 MHz (space-to-Earth), and the band 256, which comprises a pair of frequency bands: 1980–2010 MHz (Earth-to-space) and 2170–2200 MHz (space-to-Earth).

In August 2022, SpaceX and T-Mobile unveiled a groundbreaking partnership, with plans to offer direct-to-device (D2D) services in the United States, even in the most remote and previously unreachable areas, where traditional cellular signals struggle to reach. The concept revolves around creating a novel network that harnesses the capabilities of Starlink satellites and T-Mobile’s mid-band 5G spectrum. Subsequently, it was announced that Starlink’s second-generation system would provide D2D services within the Frequency Division Duplex (FDD) pair of the n25 band, specifically within the 1910–1915 MHz (Earth-to-space) and 1990–1995 MHz (space-to-Earth) bands. Starlink has further extended its global reach by partnering with cellular providers, including Rogers in Canada, Optus in Australia, One NZ in New Zealand, KDDI in Japan, and Salt in Switzerland.

It is important to note that if both bands are utilized for D2D services, the NTN satellite receivers of different Mobile Network Operators (MNOs) will be susceptible to interference from Starlink satellite transmitters. Since space-to-space interference scenarios have low attenuation of the interference signals, it is difficult to mitigate such interferences, especially taking into account that the victim receiver satellite will have in its line-of-sight range a very large number of satellites that will serve different countries, which means that it would be very difficult to manage such interference. Figure 1 illustrates the interference scenario between a typical NTN system operating in the n256 band and Starlink Generation 2, which operates in the n25 band.

The simulation characteristics of the NTN satellite can be derived from the ITU-R Report M.2514 [10] and Technical Report 3GPP TR 38.821. The spacecraft is located in low-Earth orbit (LEO). These parameters can be applied to both terrestrial and satellite spectrums for the S-band. The NTN system characteristics that were simulated in the study are presented in in Table 1.

The NTN satellite employs a multibeam antenna system for both reception and transmission. This advanced antenna comprises 19 individual beams, each designed to operate within distinct frequency ranges. This particular antenna configuration is comprehensively described in the 3GPP TR 38.821 document. Figure 2 illustrates the spatial layout of these beams on the NTN satellite’s onboard antenna.

In our analysis, we’ve considered the utilization of a typical unmodified handset (smartphone) as a reference device. Table 2 provides an overview of the smartphone’s typical characteristics.

Interference from Starlink to MSSs is expected to be most common in regions where countries using MSS equipment are geographically close to those adopting Starlink’s n25 band for direct-to-device connectivity. SpaceX has initially planned to employ the n25 band in collaboration with T-Mobile in North America. However, considering the global reach of the Starlink satellite system and the fact that the n25 band is supported for smartphones worldwide, it is likely that expansion into other regions will be on the horizon. This expansion will be especially significant given the band’s global compatibility with smartphones across all regions. Therefore, in our study, we consider interference to MSS user equipment, which is located in the desert area of African continent where satellite services will likely to be used since such remote areas do not have terrestrial networks coverage. Figure 3 illustrates the interference scenario between the MSS system operating in the 1980–2010 MHz band and Starlink Generation 2, which operates in the n25 band.

The characteristics of the MSS system were derived from the ITU-R filings. The MSS system characteristics that were simulated in the study are presented in in Table 3.

For the Starlink satellite system, we gathered simulation data from the Federal Communications Commission (FCC) records and applications. Specifically, our simulation involved 1694 Starlink satellites. It is worth noting that in practical deployments, the number of satellites could be considerably larger, as Starlink has announced plans for more extensive constellations. The simulation of Starlink satellites closely adhered to the specifications provided to the FCC.

Table 4 outlines the simulation parameters employed for modeling the Starlink system.

For both Starlink and NTN, each beam of the multibeam antenna used the pattern that was based on Recommendation ITU-R S.1528 [11], which describes typical antenna patterns for NGSO satellites in the frequency ranges below 30 GHz; these antenna patterns are commonly used for compatibility studies in the ITU-R study groups and for frequency coordination between the satellite systems [12]. While it is important to acknowledge that the antenna patterns described in the recommendation are essentially approximations, real-world tests and measurements have consistently demonstrated their validity and applicability. These patterns, despite being theoretical representations, have been proven to closely align with the actual performance of NGSO satellite communication systems in practice. In other words, they serve as reliable models that accurately represent the behavior of the antennas in real-life scenarios. The antenna pattern of the MSS is based on Recommendation ITU-R S.672, which is stated in the ITU filings of the system [13].

The antenna patterns are presented in Figure 4.

## 4. Methodology of Simulations

The study employed a hybrid approach, combining deterministic analysis and Monte Carlo simulations. The deterministic aspect involved calculating the orbital positions of the satellites based on Kepler’s laws, while the Monte Carlo simulations were utilized to generate interfering transmitters from user equipment (UE). This process enabled us to estimate the cumulative interference caused by terrestrial UE on the NTN satellite receiver. For more precise results the simulations had a step size of 1 s. Due to the complex computations, they are performed in corresponding toolkits of Matlab at the same time, to understand the mathematics that lie behind these computations. Several of the most important expressions are provided to understand how the results were obtained. 

In the case of a space-to-space interference scenario, *PL* can be calculated using the free space propagation model, following Recommendation ITU-R P.525 [14]. 

The interference level from the i-th interfering station can be calculated using the following expression:$$I=P_{interfer}+G_{interfer}\left(\varphi \right)+G_{victim}\left(\varphi \right)-PL$$
where *I* represents the interference level caused by the the i-th interfering station. *P_interferer_* denotes the output power of the interfering station, expressed in dBW. *G_interferer_* stands for the gain of the transmitting antenna on the interferer satellite, directed to the victim receiver, and measured in dBi. *G_victim_* represents the gain of the receiving antenna at the victim receiver station, oriented to the interfering station, and is also measured in dBi. *PL* stands for the propagation loss between the interfering transmitter and the victim receiver, measured in dB.

To calculate the aggregate interference level from Starlink, the following expression can be used [14]:$$I_{agg}=10\log(\sum_{j=satellites}\sum_{j=beams}10^{\frac{I\left(i\right)}{10}})$$

The Carrier-to-Noise-and-Interference Ratio (C/(N + I)) for the transmission link between the satellite and UE can be derived from the Carrier-to-Noise Ratio (C/N) and Carrier-to-Interference Ratio (C/I), as indicated by the following equations [13,14]:$$C/\left(N+I\right)=-10\log\left(10^{-0.1C/N}+10^{-0.1C/I}\right)$$

The formula for C/N calculation is [15,16]
C/N=EIRP+G/T−k−PL−B
where *EIRP* stands for effective isotropic radiated power (EIRP), expressed in dBW, *G*/*T* is the antenna-gain-to-noise-temperature, measured in dB, k is the Boltzmann constant and equals to 228.6 dBW/K/Hz, *PL* is the free space pathloss, and *B* is channel bandwidth in dBHz.

The antenna-gain-to-noise-temperature G/T can be derived by the following equation [16]:$$G/T=G_{R}+N_{f}-10\log\left(T_{0}+\left(T_{r}-T_{0}\right)10^{-0.1Nf}\right)$$
where *G_R_* is the receiver antenna gain, *N_f_* is the noise figure, *T*_0_ is the environment temperature, and *T_r_* is the receiver.

The pathloss between the interfering Starlink satellites and victim MSS/NTN satellites can be calculated using the following traditional free space expression [17]:$$PL=32.4+20\log\left(f\right)+20\log\left(d\right)$$
where *f* represents the frequency of the transmitter and *d* represents the distance between an interfering transmitter and a victim receiver.

### 4.1. Methodology of the Simulation of Interference to MSS

It should be noted that most MSS systems use a transparent payload. Users transmit within the frequency range of 1980–2010 MHz, subsequently, the frequency is converted to the 6700–7075 MHz range before retransmission to the Earth station gateway, which actively tracks the MSS system. Since there is no onboard processing, it is required to calculate the composite C/(N + I) of the transparent payload to estimate end-to-end performance of the link. The following expression can be used to calculate end-to-end performance [15,16]:$$C/{\left(N+I\right)}_{total}=-10\log\left(10^{-0.1C/{\left(N+I\right)}_{up}}+10^{-0.1C/{\left(N+I\right)}_{down}}\right)$$
where *C*/(*N* + *I*)*_up_* is the signal-to-noise ratio plus interference level in the Earth-to-space link, and *C*/(*N* + *I*)*_down_* is the signal-to-noise ratio plus interference level in the space-to-Earth link [18].

Figure 3 shows the simulation of the end-to-end performance of the MSS system while the interfering Starlink satellite is in proximity to it.

After C/N and C/(N + I) were calculated, it is possible to calculate the Eb/No. The Eb/No is directly related to the C/N value, and can be expressed as follows [19]:$$E_{b}/N_{0}=C/N-10\;\log\left(R/B\right)$$
where *E_b_*/*N*_0_ represents the ratio of energy per bit to spectral power density (dB); *N* is the noise level in a reference bandwidth (dBW); *I* represents the interference level in a reference bandwidth (dBW); *R* is the data rate (kbit/s); *B* is the reference bandwidth (kHz); and *C* is the carrier bandwidth [20].

The total noise level under interference conditions should be presented as a sum of the spectral density of the receiver’s noise and external noise, N_∑_ = N_o_ + I_o_. The levels of E_b_/(N_o_ + I_o_) can be checked according to the curves to calculate the BER levels depending on the modem implementation. In our study we have considered the most commonly used modulation scheme QPSK. Since there are different modem implementations for MSS no code rate was considered and raw BER levels were obtained. The threshold BER levels depend on the system and usually varies; however, the most common threshold BER level is 10^−6^. A lot of MSS systems employ this threshold in their specifications, including the Globalstar satellite system. Figure 5 shows a simulation of interference from Starlink to the MSS system.

### 4.2. Methodology of the Simulation of Interference to NTN

In the realm of satellite communications, the evaluation of throughput losses involves the examination of Modulation and Coding Schemes (MODCOD). These coding schemes are typically outlined in corresponding specifications. However, when it comes to incorporating adaptive modulation and coding schemes into simulations, there exists no universally accepted methodology.

In contrast, 3GPP terrestrial specifications have made significant progress in this area. They have developed a methodology that effectively accommodates adaptive modulation and coding schemes. To achieve this, 3GPP has introduced link adaption approximations [21], which enable the accurate estimation of modem throughput losses, taking into account the intricacies of adaptive modulation and coding schemes.

Given that NTN system D2D systems are designed to use the same waveforms to ensure compatibility with smartphones, these link-level adaptions developed by 3GPP can be readily applied to the analysis of interference within the NTN system.

The subsequent equations serve to approximate throughput over a channel, given a specific signal-to-interference-plus-noise ratio (SINR), measured in dB, when employing link adaptation:$$Throughput\;\left(SINR\right),\;bps/Hz=\left\{\begin{array}{cc} \;\;\;\;0 & for\;SINR<SINR_{MIN}\\ \alpha \cdot S\left(SINR\right) & for\;SINR_{MIN}\leq SINR<SINR_{MAX}\\ \alpha \cdot S\left(SINR_{MAX}\right) & for\;SINR\;\geq \;SINR_{MAX} \end{array}\right.$$
where:*S*(*SINR*): Shannon bound, *S*(*SINR*) = log_2_(1 + 10*^SINR^*^/10^) (bps/Hz);α: Attenuation factor, representing modem implementation losses;*SINR_MIN_*: Minimum SINR of the code set, dB;*SINR_MAX_*: Maximum SINR of the code set, dB.

In these equations, the Shannon bound represents the maximum theoretical throughput than can be achieved over an AWGN channel for a given SNIR. The parameters α, *SINR_MIN_*, and *SINR_MAX_* can be chosen to represent different modem implementations and link conditions [22]. The parameters from Table 5 represent a typical case, which assumes the following:1:1 configuration of the antenna;AWGN channel model; Link Adaptation;No HARQ.

Based on the above equations, bitrate mapping can be calculated for the uplink and downlink. Figure 6 represents bitrate mappings for the downlink and uplink of the typical NTN system.

To determine the throughput loss for both the uplink and downlink in a considered network, the essential steps involve calculating the signal-to-noise ratio (SNR) of all the connections and assessing interference (I) from interfering stations. Next, the interference level should be combined with the background noise level in the analyzed system to compute the signal-to-interference-plus-noise ratio (SINR). These SINR values can then be compared against the reference curves in Figure 6, and the throughput loss can be determined using the following formula:$$\mathrm{Throughput}\left[kbps\right]=\frac{N_{RB\_per\_UE}}{N_{total\_RBs}}\cdot S_{capacity}\;\left(SINR\right)\cdot B\cdot 1000$$
where *Throughput*[*kbps*] is the maximum throughput of the channel, expressed in bps, *N_RB_per_UE_* is the number of resource blocks (RBs) per user, *N_total_RBs_* the total number of RBs, *B* stands for the channel bandwidth, expressed in MHz, and *S_capacity_* is the spectral efficiency, depending on the *SINR*, expressed in bps/Hz.

Figure 7 illustrates a simulation depicting interference from a Starlink satellite to the NTN satellite receiver when the NTN satellite is serving a user during a close flyby of the Starlink satellite for a single-entry interference case and for aggregate interference.

## 5. Simulation Results

The findings of our studies are presented through Cumulative Distribution Functions (CDF) illustrating the reduction in carrier-to-noise levels and the resulting throughput loss in the NTN satellite uplink and BER rise of the MSS composite link. These metrics allow us to effectively estimate the real degradation of the MSS and NTN systems’ operations when they are interfered with by the Starlink systems in the 1990–1995 MHz band. 

It is worth noting that, in Recommendation ITU-R S.2131satellite communications systems typically consider a threshold C/N reduction of 1 dB, corresponding to a 10% reduction in spectral efficiency [23]; this protection criterion can be applied to MSS systems. 

In accordance with 3GPP specifications for terrestrial LTE/NR segments, the threshold for acceptable throughput loss stands at 5% [22]. This protection criterion can be applied be applied to the satellite NTN system offering D2D services.

### 5.1. Single-Entry Interference from Starlink to MSS

In this scenario, the interfering Starlink satellite was in close proximity to the MSS system, which was actively receiving a transmission from user equipment. This scenario lasted approximately 200 s, aligning with the MSS uplink access duration while considering the necessary carrier-to-noise values. The average distance between the victim MSS receiver and the Starlink transmitter was 1600 km. Figure 8 displays the CDFs of single-entry interference from the Starlink satellite operating in the 1990–1995 MHz frequency band to the typical MSS system.

The results indicate that in the case of single-entry interference, specifically in a scenario involving the proximity of a Starlink satellite to an MSS system, the end-to-end link performance of the MSS system can experience an Eb/No degradation from 1 to 3 dB. For the most part, the overall BER levels are within acceptable limits. However, it is noteworthy that there can be a significant surge in BER levels under certain conditions. Considering that our MSS link simulation assumed close to ideal conditions, it is reasonable to speculate that in practical scenarios with this type of interference, BER levels could exceed acceptable thresholds for a substantial portion of the time.

### 5.2. Aggregate Interference from Starlink to MSS

In this scenario, we considered the aggregate interference level originating from the Starlink constellation, which comprised 1694 interfering satellites. Starlink satellites were active only when located above landmass areas and remained inactive over oceanic regions. Figure 9 illustrates the results of the aggregate interference from the Starlink system operating in the 1990–1995 MHz frequency band to the typical MSS system.

The results indicate that in the case of aggregate interference, the MSS end-to-end link experiences a degradation exceeding 10 dB for a significant duration, far surpassing the acceptable threshold for a tolerable Eb/No degradation. This leads to a significant increase in BER levels. The elevated interference BER levels pose a serious threat to the complete outage of MSS uplink operations within the 1990–1995 MHz frequency range. As a result, once the second-generation Starlink system is deployed and serves millions of users, it will render it practically impossible to deploy MSS systemss in specific areas operating within the 1990–1995 MHz frequency band.

### 5.3. Single-Entry Interference from Starlink to NTN

Within this scenario, we estimated interference levels and throughput losses over several minutes as the NTN satellite served a user while an interfering Starlink satellite passed nearby. The average distance between the NTN satellite and the Starlink satellite was approximately 1900 km. Our results are presented in CDF curves illustrating the levels of signal-to-noise degradation and the associated throughput losses. Figure 10 displays the CDFs of the single-entry interference from the Starlink satellite operating in the 1990–1995 MHz frequency band to the NTN system.

Our results indicate that, in the case of single-entry interference, the throughput loss amounts to 2.5%. Generally, a 5% throughput loss is considered acceptable. However, this criterion is typically applied to terrestrial LTE/NR systems. Given the NTN satellite’s limited uplink link budget for D2D services, even a 2.5% throughput loss can substantially impact the performance of the NTN system.

### 5.4. Aggregate Interference from Starlink to NTN

In this scenario, we considered the aggregate interference level originating from the Starlink constellation, which comprised 1694 interfering satellites. Starlink satellites were active only when located above landmass areas and remained inactive over oceanic regions. It is important to note that, since the NTN satellite utilizes nineteen spot beams, we assumed that only seven beams directly overlap with the Starlink interferers, while the other twelve beams receive out-of-band interference. Figure 11 illustrates the results of the aggregate interference from the Starlink system operating in the 1990–1995 MHz frequency band.

Our findings reveal a significant throughput loss in the case of aggregate interference, amounting to 40%. This throughput loss greatly exceeds the 5% threshold and would lead to a substantial degradation in the quality of service provided by the future NTN systems operating in the n256 band. This would result in a highly inefficient spectrum utilization in the uplink channel of the 1980–2010 MHz band.

## 6. Conclusions

The evolution of non-terrestrial network (NTN) systems for direct-to-device (D2D) services brings significant advantages, leveraging unmodified handsets for seamless integration. However, the competitive pursuit of frequency bands compatible with smartphones has led to potential interference issues. Specifically, the overlap of Starlink in frequency band 1990–1995 MHz with existing MSS systems and future NTN systems in the n256 band in, poses challenges with space-to-space link interference. Our study demonstrates that this interference can have severe implications; specifically, our results indicate up to a 40% reduction in uplink throughput for future NTN systems and a significant degradation of the services for the currently existing MSS systems.

It is essential to address this issue, given its significant impact on users and the potential depletion of valuable spectrum resources. Considering the inherent challenge of avoiding space-to-space interference, especially given the anticipated high number of active Starlink satellites, there are only two feasible solutions for interference mitigation. The first involves using non-overlapping bands within the n25 band. The second entails separating the MSS/NTN and Starlink service areas. This separation can be implemented by countries aiming to provide satellite services within their territories, requiring explicit agreements and coordination between different administrations to prevent mutual interference.

## Figures and Tables

**Figure 1 sensors-24-01297-f001:**
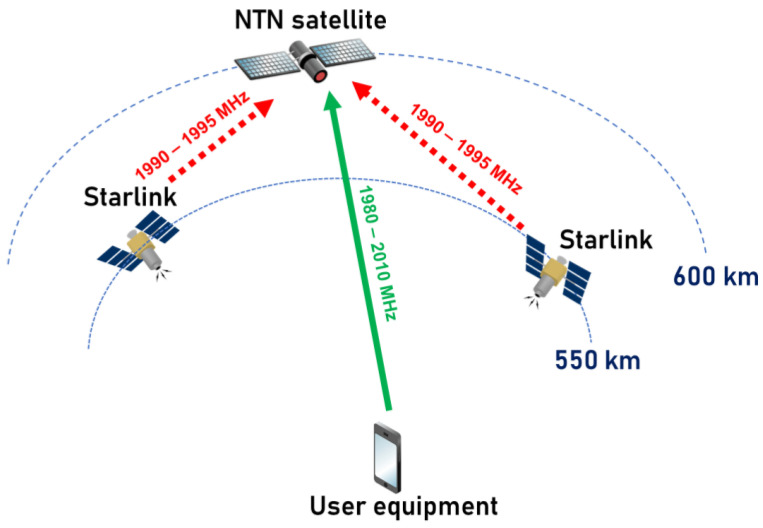
Interference scenario between NTN network operating in the 1980–2010 MHz and Starlink operating in the 1990–1995 MHz.

**Figure 2 sensors-24-01297-f002:**
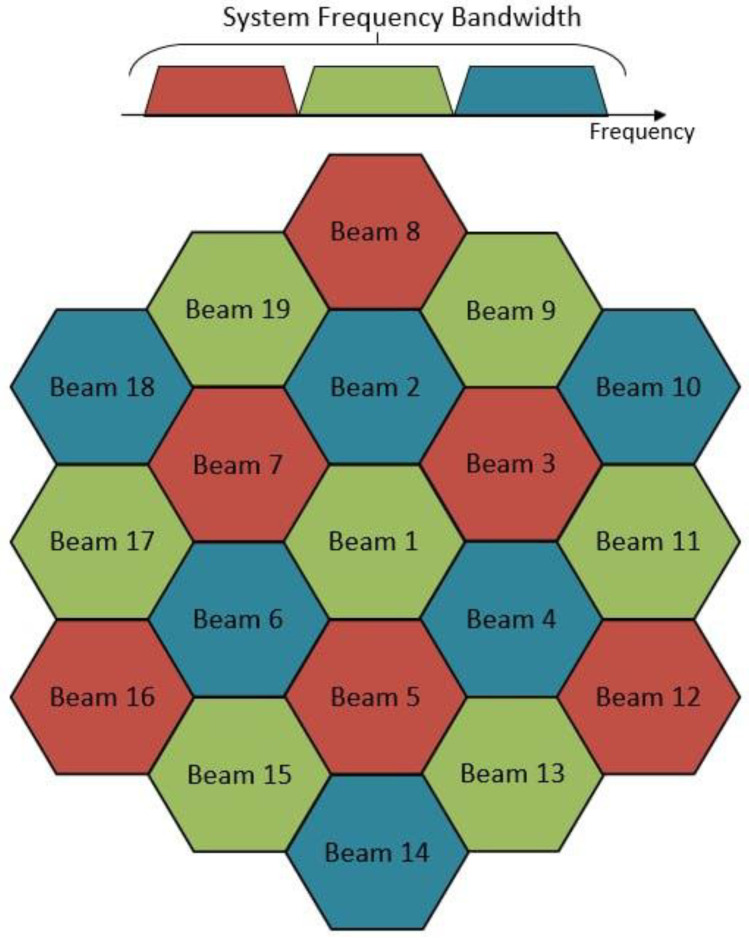
Illustration of the allocation of the beams of the NTN satellite used in simulations.

**Figure 3 sensors-24-01297-f003:**
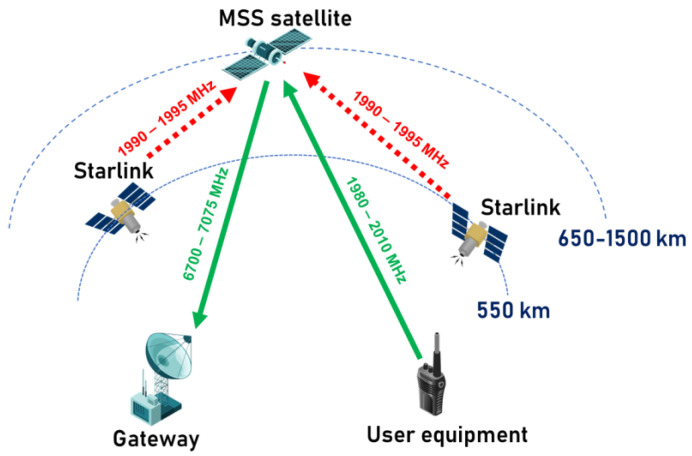
Interference scenario between NTN users and terrestrial users.

**Figure 4 sensors-24-01297-f004:**
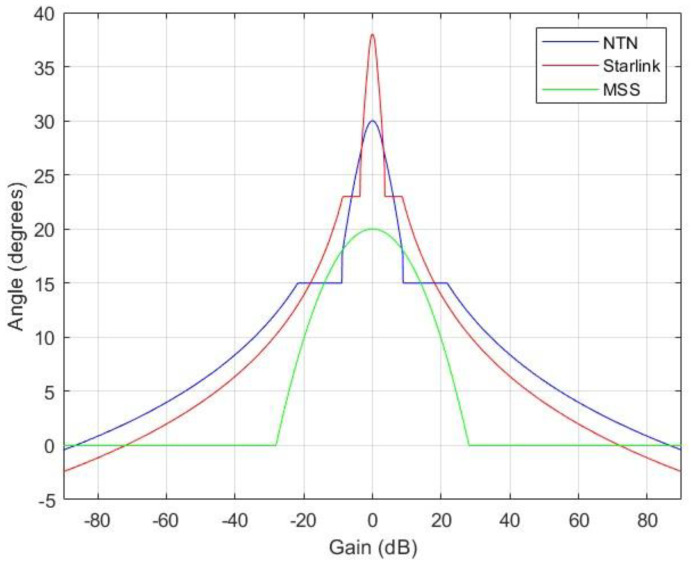
Antenna pattern approximations of NTN satellite and Starlink used in the simulations.

**Figure 5 sensors-24-01297-f005:**
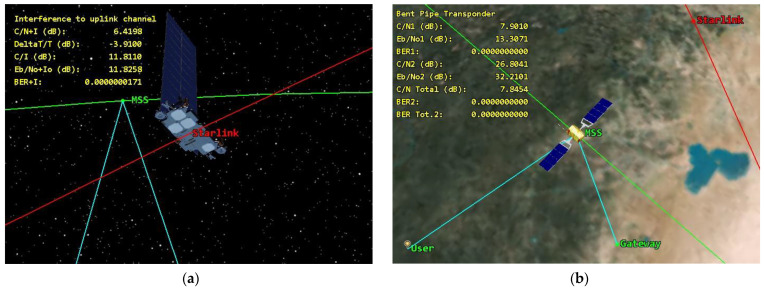
Simulation of the single-entry interference from a Starlink satellite to the MSS receiver: (**a**) view from Starlink; (**b**) view from MSS.

**Figure 6 sensors-24-01297-f006:**
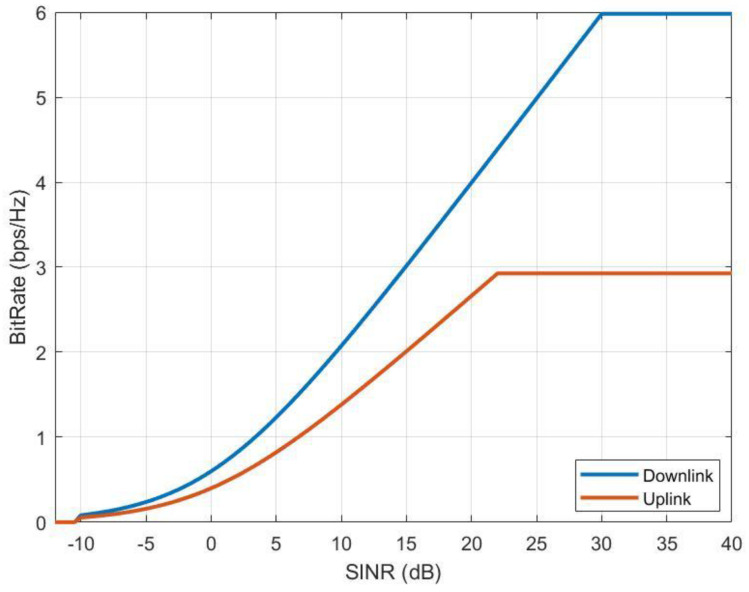
Bitrate mappings of the NR downlink and uplink.

**Figure 7 sensors-24-01297-f007:**
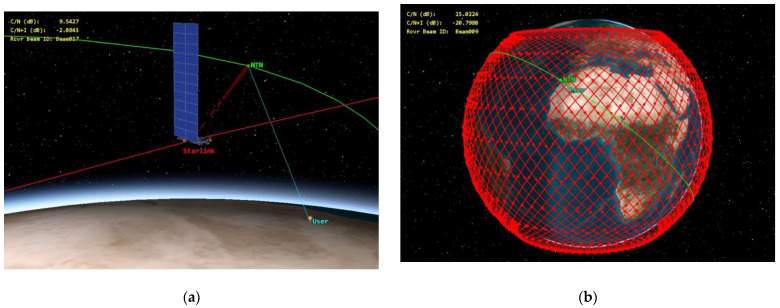
Simulation of the interference from a Starlink satellite to the NTN satellite receiver: (**a**) single-entry interference; (**b**) aggregate interference.

**Figure 8 sensors-24-01297-f008:**
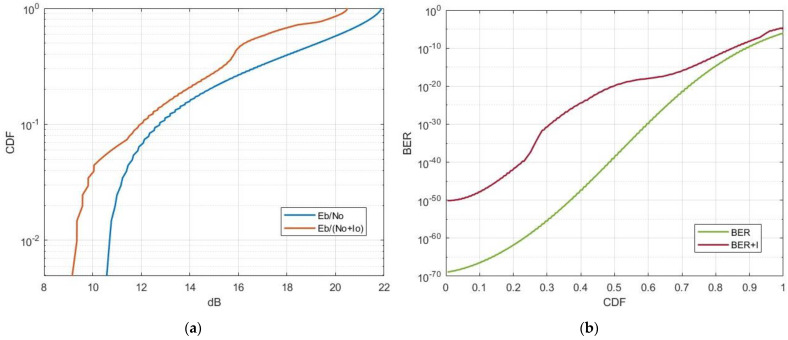
CDF of single-entry interference from Starlink to the NTN satellite receiver: (**a**) carrier-to-interference reduction; (**b**) throughput loss of the NTN uplink.

**Figure 9 sensors-24-01297-f009:**
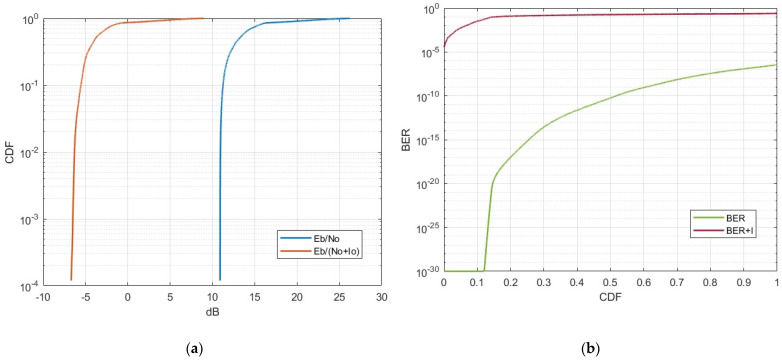
CDF of aggregate interference from Starlink to the MSS satellite receiver: (**a**) carrier-to-interference reduction; (**b**) throughput loss of the NTN uplink.

**Figure 10 sensors-24-01297-f010:**
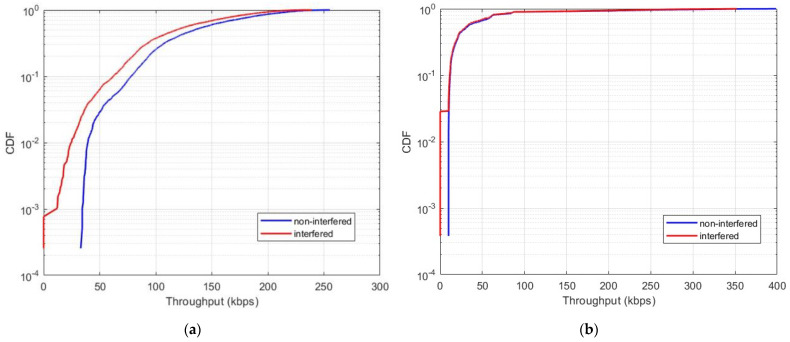
CDF of single-entry interference from Starlink to the NTN satellite receiver: (**a**) carrier-to-interference reduction; (**b**) throughput loss of the NTN uplink.

**Figure 11 sensors-24-01297-f011:**
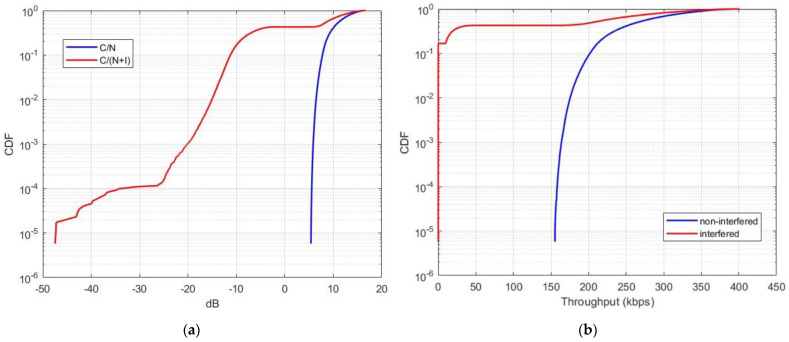
CDF of aggregate interference from Starlink to the NTN satellite receiver: (**a**) carrier-to-interference reduction; (**b**) throughput loss of the NTN uplink.

**Table 1 sensors-24-01297-t001:** Simulation parameters of NTN satellite.

Parameter	Value
Orbit height	600 km
Multibeam pattern	At least 19 beams of the hexagonal form
Frequency	S-band
System bandwidth	30 MHz
Channel bandwidth	Space-to-Earth: 30 MHz divided by frequency reuse factor Earth-to-space: 180 kHz
Beamwidth	4.41 degrees
EIRP	34 dBW/MHz
Maximum antenna gain	30 dBi
Satellite G/T	1.1 dB/K
Traffic model	Full buffer
UE antenna height	1.5 m

**Table 2 sensors-24-01297-t002:** Handset characteristics that were used in the simulations.

Parameter	Value
Antenna type and configuration	Omni-directional
Polarization	Linear: ±45° X-pol
Antenna gain (dBi)	0
Antenna temperature (K)	290
Noise figure (dB)	7
Tx transmit power	200 mW (23 dBm)

**Table 3 sensors-24-01297-t003:** Simulation parameters of the MSS system.

Parameter	Value
Orbit height	650–1500 km
Inclination angle	82–96 deg
Channel bandwidth	25 KHz
User terminal antenna pattern	Omni
EIRP of user terminal	5–10 dBW
Satellite antenna gain	20 dBi
Gateway antenna gain	48 dBi
Modulation	QPSK

**Table 4 sensors-24-01297-t004:** Simulation parameters of the Starlink system.

Parameter	Rural-eMBB-s
Orbit height	550 km
Inclination angle	53 deg
Number of planes	72
Number of satellites per plane	22
EIRP spectral density	−2.3 dBW/Hz
Satellite max antenna gain	38 dBi
Spectral power density	−40.3 dBW/Hz
Max input power	19.7 dBW/MHz
Out-of-band attenuation	56 dB
Out-of-band power	−36.3 dBW/MHz

**Table 5 sensors-24-01297-t005:** Link adaptation parameters of NR.

Parameter	Downlink	Uplink	Notes
α	0.6	0.4	Represents implementation losses
*SINR_MIN_*, dB	−10	−10	Based on QPSK, 1/8 rate (DL) & 1/5 rate (UL)
*SINR_MAX_*, dB	30	22	Based on 256-QAM, 0.93 rate (DL) & 64-QAM, 0.93 rate (UL)

## Data Availability

Data is contained within the article.

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
