# Peer review of "Exploring Interference Issues in the Case of n25 Band Implementation for 5G/LTE Direct-to-Device NTN Services"

_sensors, 2024, doi:10.3390/s24041297_

Round 1
Reviewer 1 Report
Comments and Suggestions for Authors
This manuscript presents a thorough analysis of the potential interference challenges posed by the implementation of 5G/LTE Direct-to-Device NTN Services in the n25 band. The authors provide a comprehensive assessment of these challenges, particularly in the context of the Starlink generation 2 satellites and Gonets-M1 mobile satellite system. The methodology combining deterministic analysis with Monte-Carlo simulations is robust, and the presentation of results through Cumulative Distribution Functions is clear and informative.
Author Response
OK, thank you for the comments!
Reviewer 2 Report
Comments and Suggestions for Authors
The widespread adoption of mobile technologies in today's society has revolutionized the way people stay connected and access information. With each new generation of technology emerging roughly every decade, we have witnessed a remarkable increase in data rates. Currently, the industry has reached a saturation point where users generally have sufficient data rates for their everyday needs. Consequently, the primary focus has shifted towards meeting the growing demand for service availability, rather than constantly seeking higher data rates.
The evolution of Non-Terrestrial Network (NTN) systems for Direct-to-Device (D2D) services offers a multitude of advantages compared to other mobile or fixed satellite systems. Notably, it enables communication services on the very same unmodified handsets that users employ for terrestrial services. This removes the need for specialized subscriber terminals, as users can exploit the full potential of their smartphones for seamless integration. The advent of D2D communication systems has ignited a competitive race for securing frequency bands compatible with smartphones. However, many satellite operators so far select these bands without due consideration for potential interference issues. Some terrestrial bands overlap with MSS and NTN systems. For example, in the case of Starlink, the 1990-1995 MHz frequency band, part of the n25 band, overlaps with the MSS systems and with the n256 band (1980-2010 MHz) planned for NTN services. One major problem is that 1990-1995 MHz operates in the space-to-Earth direction, while 1980-2010 MHz operates in the Earth-to-space direction, leading to space-to-space link interference that is challenging to mitigate.
Identifying frequency bands with minimal interference is of paramount importance, as interference can impose additional constraints on users and disrupt the quality of service. The frequency bands 1910-1915 MHz (Earth-to-space) and 1990-1995 MHz (space-to-Earth), chosen by SpaceX for direct-to-device services, pose a significant challenge. The Earth-to-space segment overlaps with the mobile satellite service band 1980-2010 MHz, which is used in the reverse direction (Earth-to-space). Furthermore, this band is a part of the 3GPP n256 NTN band.
This situation creates a notable problem due to space-to-space interference between Starlink and other MSS and NTN systems. For MSS using the Gonets-M1 system as an example, which will offer voice, data transfer, and IoT services in the future, we have demonstrated that a comprehensive implementation of the 1990-1995 MHz band by the 2nd generation Starlink satellites will result in persistent and unacceptable interference in many regions. Additionally, our study indicates that if NTN systems will be deployed in the n256 band, the interference from Starlink will potentially result up to 40% reduction in the uplink throughput of the n256 band. This substantial reduction poses a critical concern for NTN systems. This interference would render the 5 MHz portion of the 1980-2010 MHz band unusable for other MSS and NTN systems, significantly depleting this valuable spectrum resource.
Potential solutions for this problem may involve Starlink's utilization of non-overlapping portions within the n25 band or territorial separation strategies. However, the latter may be complicated due to the high intensity of space-to-space interference even among satellites serving different territories. Thus, system engineers may consider the findings of this study as a crucial factor in developing hardware-level interference mitigation solutions, whereas spectrum regulators of different countries need to proceed with caution when granting approval for the operation of the Starlink system in the n25 frequency band within their territories.
Comments on the Quality of English LanguageMinor editing of English language required
Author Response
Thank you for your comments! We made some editorial changes in the text, trying to address the issue of English editing.
Reviewer 3 Report
Comments and Suggestions for Authors
Please address the following points in the manuscript.
1. Please mention the name of the toolkit/software used for the simulation analysis of the proposed work.
2. In section 3.1, Line 289, “R is the data rate (kbits/s), B is the reference bandwidth (KHz)”, please highlight what this reference bandwidth is.
3. In section 4.1, line 385, please mention clearly what’s the performance metric that is degraded by 1 to 3 dB. Is this the C/N or C(N+I)? The same comment also applies to section 4.2, Line 403.
4. Please discuss the BER curves provided in Figure 8B, are coded BER or Uncoded?
5. Please correct the numbering of the sections/subsections, e.g. Simulation results are numbered 5 while there are subsections which are numbered 4.1, 4.2, 4.3, and 4.4. The same comment also applies to section 4 where subsections are numbered in 3.x.
6. Why the subsections 4.1, 4.2, 4.3 and 4.4 are named the same. Also, Throughput and BER for each scenario can be combined in one subsection (e.g. combine sections 4.1 & 4.3 and sections 4.2 & 4.4).
7. The authors mentioned in section 4.2, line 402, “ The results indicate that in the case of aggregate interference, the Gonets-M1…………, far surpassing the acceptable threshold for tolerable degradation.” Please indicate the acceptable threshold in the plot in Figure 9 and also mention it in the manuscript in the text. Also, please comment – as the BER curve is surpassing the acceptable threshold for tolerable degradation, what about the Block Error Rate (BLER) which is considered practically in a communication system? Please provide any reference in the literature in section 2, if any such study has been performed where BLER performance is investigated instead of BER.
8. Please also explain in the manuscript, how can be dealt with these types of interferences. It would be great if a solution is also proposed clearly in the manuscript.
9. The conclusion is too lengthy, please rewrite a brief conclusion of the proposed work.
Comments on the Quality of English LanguageThe quality of the English is fine but can be improved further.
Author Response
Thank you for your comments!
- OK, we added that we used Matlab for simulations.
- The reference bandwidth is the carrier bandwidth, we have added that clarification
- Eb/No has degraded and Eb/No+Io is the level of Eb/No after the degradation
- Added the information that it was uncoded
- Thank you for the observation, corrected
- We have renamed them, these subsections represent results for single-entry and aggregate MSS and NTN, so there are 4 subsections
- Typically the accepted BER is 10^-6 for example it's the accepted BER for Globalstar, in other systems it varies, but 10^-6 is the most common one, we added this information
- We added how it can be solved in the conclusions section, overall it can be solved only either by frequency separation or explicit agreements by the countries so the systems would operate only above the concrete regions/territories
- We have shortened the conclusion
Reviewer 4 Report
Comments and Suggestions for Authors
Introduction is not descriptive
Literature survey is not done properly, a comparative table mentioning the already existing work compared with the propsed one need to be highlighted and explained
system model is not there in the paper
novelty in the results is lacking
Comments on the Quality of English Languagegrammatical and typos need to be corrected
Author Response
Thank you for your comments! We have edited introduction, literature survey and conclusions, making it sound more comprehensible.
The novelty of these results is that this problem in general wasn't studied before extensively, and given that D2D systems are currently seeking for spectrum and already launch test satellites, we still have time for discussions on how to avoid the interference for such systems.
Round 2
Reviewer 3 Report
Comments and Suggestions for Authors
The revised manuscript is now in better shape to be accepted for publication.
Comments on the Quality of English LanguageThe English language in this manuscript is fine
Reviewer 4 Report
Comments and Suggestions for Authors
The author has modified the manuscript as per the comments